# COVID-19 Virus and Vaccination Attitudes among Healthcare Workers in Michigan: A Cross-Sectional Study

**DOI:** 10.3390/vaccines11061105

**Published:** 2023-06-16

**Authors:** Maya Asami Takagi, Samantha Hess, Karissa Gawronski, Nicholas Haddad, Bernard Noveloso, Stephen Zyzanski, Neli Ragina

**Affiliations:** 1College of Medicine, Central Michigan University, Mt. Pleasant, MI 48859, USA; takag1ma@cmich.edu (M.A.T.); hess2sm@cmich.edu (S.H.); gawro1k@cmich.edu (K.G.); hadda1ne@cmich.edu (N.H.); novel1bd@cmich.edu (B.N.); 2Department of Family Medicine and Community Health, School of Medicine, Case Western Reserve University, Cleveland, OH 44106, USA; zyzan1sj@cmich.edu

**Keywords:** COVID-19 virus, COVID-19 vaccines, vaccine hesitancy, healthcare workers, vaccination recommendations

## Abstract

Background: Defining the characteristics of healthcare worker (HCW) attitudes toward the coronavirus disease 2019 (COVID-19) vaccine can provide insights into vaccine hesitancy. This study’s goal is to determine HCWs’ attitudes regarding the COVID-19 vaccination and reasons for vaccine hesitancy. Methods: This cross-sectional study surveyed HCWs working in institutions in Saginaw, Sanilac, and Wayne counties in Michigan (N = 120) using tipping-scale questions. Analysis of variance and t-test were used to measure HCWs’ attitudes toward the COVID-19 virus and vaccines. Results: Most HCWs received (95.9%) and recommended (98.3%) a COVID-19 vaccine. The top three factors that HCWs cited for recommending a COVID-19 vaccine were: (1) efficacy of the vaccine, (2) current exposure to patients with active COVID-19 infection and risk of virus spread, and (3) safety of vaccine and long-term follow-up. Female HCWs or HCWs aged 25–54 years were more concerned about contracting COVID-19. Physicians or HCWs aged 55–64 were less concerned regarding the effectiveness and side effects of the vaccine. Conclusions: Gender, age, ethnicity, provider type, and medical specialty showed statistically significant differences among COVID-19 attitudes. Focusing educational efforts on HCW demographics who are more likely to have negative attitudes can potentially decrease vaccine hesitancy.

## 1. Introduction

Coronavirus disease 2019 (COVID-19) is a disease caused by the SARS-CoV-2 virus that was first identified in Wuhan, China, and spreads from human-to-human through airborne particles and droplets [1]. On 11 March 2020, the World Health Organization declared COVID-19 a global pandemic and labeled the virus as a public health issue warranting international concern [2]. To halt the spread of the COVID-19 virus, varying efforts were enacted globally. These methods included, but were not limited to, national stay-at-home orders, mask use in public spaces, and border controls [3,4]. However, these efforts quickly proved to not properly prevent the spread of COVID-19. Mass vaccination served as a potential solution to combat the COVID-19 pandemic in a way that other efforts had been unable to do. Although, with the presentation of a new vaccine, vaccine hesitancy quickly became a recognized impediment to slowing the COVID-19 pandemic [5]. Vaccine hesitancy makes a significant contribution to knowledge, attitudes, and practices toward pandemic relief efforts, and leads to suboptimal vaccination coverage [6,7,8]. In an attempt to remedy this barrier, collecting data that defines the characteristics of the general population’s attitudes and healthcare workers (HCWs) toward the COVID-19 vaccines could provide insight into vaccine hesitancy [6,7,8,9,10,11]. 

As preliminary research on a coronavirus vaccine was being conducted, herd immunity became the standard for gaining control of the COVID-19 pandemic [12]. Herd immunity is defined as an unlikely spread of disease from human-to-human due to pre-existing immunity to the disease in a large portion of the population through vaccination or prior illness [13]. Therefore, the concept of herd immunity brought forward a higher importance to the vaccine, with a high percentage of the general population having to become infected and risk death and other debilitating complications of the coronavirus disease as the alternative to having mass vaccination. Herd immunity through mass vaccination also has the essential benefit of the protection of individuals who cannot and/or have not yet received the vaccine [14]. Thereby, mass vaccination protects the vulnerable members of society who would be at greater risk for complications or death from COVID-19 and any hesitancy towards the vaccine will complicate the process of gaining herd immunity. 

A review summarized the findings of 74 studies that investigated COVID-19 vaccination acceptance among HCWs, illustrating that almost two-thirds of HCWs were willing to accept a COVID-19 vaccine. This review highlighted common reasons for hesitancy, such as concerns about vaccine safety and efficacy, speed of development, certain roles of HCWs, and mistrust in the public health response [15]. However, studies from different regional locations yielded conflicting results among HCWs, such as variations in vaccine acceptance, sources that participants deemed trustworthy, and demographic groups who are less likely to accept a COVID-19 vaccination [16,17,18,19,20,21,22,23]. Therefore, region-specific demographic studies may better reflect an area’s values and cultures, and local public health leaders can use this knowledge to better address issues on a targeted basis.

Even in the face of growing vaccine hesitancy, HCWs still remain the most important source of information regarding vaccines and their efficacy to the general public [24,25]. The opinions of HCWs on vaccinations play an essential role in slowing the spread of COVID-19 and increasing the likelihood of positive outcomes for patients during the pandemic. This study serves as a resource to report the level of COVID-19 vaccination acceptance, recognize the top factors for recommending a COVID-19 vaccine, identify demographic characteristics associated with negative beliefs regarding COVID-19, and further extrapolate COVID-19 vaccination views among HCWs across multiple healthcare centers in Michigan. This is of great importance, as studying vaccination perception in HCWs is essential for protecting patient safety, promoting occupational health and well-being, building public trust, developing evidence-based policies, and enhancing outbreak response capabilities. It enables healthcare organizations and policymakers to address barriers, dispel myths, and implement targeted interventions that support vaccination uptake among this crucial group of professionals. By identifying the characteristics and demographics of HCWs that provide further insight into their level of vaccine acceptance or hesitancy, targeted solutions can be enacted to combat vaccine hesitancy. Therefore, focusing educational efforts on HCW demographics that have shown a higher likelihood of negative attitudes towards the COVID-19 vaccine can potentially decrease their vaccine hesitancy, and hopefully, as a result, decrease the vaccine hesitancy of their patients. Reflecting on the hesitations of vaccination during public health crises and having a descriptive understanding of negative views will better aid in addressing hesitancy in future pandemics.

## 2. Materials and Methods

### 2.1. Study Design 

This cross-sectional, pilot study collected data using questionnaires at healthcare institutions in Michigan. This study was conducted from 12 July 2021 to 30 November 2021 (Figure 1 and Figure 2) [26]. Research assistants recruited participants via institution-affiliated clinician email listservs. This study utilized one questionnaire aimed at understanding the perceived attitudes of participants regarding SARS-CoV-2 and the COVID-19 vaccines. The participants completed the questionnaire online after informed consent was obtained. The informed consent informed participants that the data recorded from this study would remain anonymous and there was no patient-identifying information gathered during this study. Participants were able to continue to the next question of the questionnaire even if they failed to provide a response to an item. Participants did not receive any gifts and were not monetarily compensated. Questionnaires were distributed by Collaborative Institutional Training Initiative (CITI)-trained Central Michigan University (CMU) College of Medicine students. The CMU College of Medicine Research Institutional Review Board (IRB), Covenant Medical Center IRB, and Saint Joseph Mercy Health System and Trinity Health System Level Research IRB provided approval and oversight to maintain ethical standards and participant anonymity. Prior to data collection, written consent to conduct the study was obtained from community affiliations partnered with the CMU College of Medicine at the locations where questionnaires were administered.

### 2.2. Participants 

Participants were recruited from outpatient clinics, an academic hospital, and a specialty clinic in four counties throughout Michigan: Isabella, Saginaw, Sanilac, and Wayne counties in Michigan (Appendix A). This included one outpatient clinic and one specialty clinic in Isabella County, three clinics and one academic hospital in Saginaw County, two clinics in Sanilac County, and one clinic in Wayne County. These healthcare institutions were selected since they employed HCWs who served the highest number of patients in those respective counties. A total of 403 HCWs were recruited via email from institution-affiliated clinician listservs. A total of 120 HCWs who resided in 16 Michigan counties completed the survey (30.3% response rate). The inclusion criteria were defined as a clinician at one of the previously mentioned healthcare institutions who had clinical certifications and/or medical licensure, was above the age of 18 years, and was able to understand English. The HCWs recruited for this study included physicians, nurses, nurse practitioners, occupational and physical therapists, medical assistants, and pharmacists from all healthcare disciplines. 

### 2.3. Measures 

A 45-item anonymous online-based questionnaire was distributed to the HCWs (Appendix A). The surveys were collected via the Qualtrics online survey platform between 12 July 2021 and 30 November 2021. The questionnaire obtained information on the following domains: demographics, the likelihood of receiving the COVID-19 vaccination, vaccination status, COVID-19 virus and vaccines beliefs and concerns, and the likelihood of recommending a COVID-19 vaccination to patients. Questions regarding demographics, virus and vaccine knowledge, and vaccination status consisted of multiple-choice answers. All questions were optional to complete. Of the 45-item questionnaire, 29 of these utilized a 3-point Likert scale which included 2 = agree, 1 = unsure, or 0 = disagree to assess attitudes regarding the COVID-19 virus and vaccine. Finally, the participants were asked to rank seven factors in order of importance for recommending a COVID-19 vaccine to a patient: efficacy of the vaccines, safety of the vaccines and long-term follow-up, duration of protection by the vaccines, the incidence of major and minor adverse effects, recommendation of the vaccines by political officials, recommendation of the vaccines by healthcare authorities, and current exposure to patients with active COVID-19 infection and risk of current spread. 

### 2.4. Statistical Analysis 

A sample size of N = 120 is estimated to have 80% power at a two-tailed 0.05 level of significance to detect a medium effect size (half a standard deviation difference among subgroup means). Independent two-sample t-tests and analysis of variance were used to examine the HCWs’ demographics and their attitudes toward the COVID-19 virus and vaccines. We used the Statistical Package for Social Sciences (SPSS) for all analyses reported in this study.

## 3. Results

### 3.1. Clinician Characteristics

The cohort consisted of 120 HCWs residing in 16 counties in Michigan. The demographic profile presented in Table 1 shows the clinician respondents were mostly White, female, and over the age of 35 years. The majority were primary care physicians who had been tested for COVID-19, received the vaccine, and would recommend the vaccine to their patients. Moreover, the vast majority of these clinicians also received their flu shots last year as well as this year.

### 3.2. Attitudes towards the COVID-19 Virus and Vaccines

Literature-validated [27,28,29,30,31,32,33,34,35] 29-item tipping scale statements related to COVID-19 virus and vaccines beliefs and concerns were used to gauge attitudes towards the COVID-19 virus and vaccines. Table 2, Table 3, Table 4, Table 5, Table 6 and Table 7 represent mean item scores of the HCWs regarding their concerns and beliefs of the COVID-19 virus and vaccines. A value closer to ‘0′ indicates that the demographic group disagreed more and a value closer to ‘2′ indicates that the group agreed more with the statements abbreviated in each column.

When comparing the responses between female and male HCWs (Table 2), female respondents were significantly more concerned about being infected by the virus than the male respondents. Female HCWs showed a greater agreement trend that the Centers for Disease Control (CDC) recommendations are more effective against the virus than male respondents. In addition, female HCWs showed a trend towards being more concerned about long term effects of the vaccine compared to males. Overall, both genders reported knowing whom to trust for COVID information. However, more males than females tended to report finding it hard to know whom to trust for COVID-19 information.

When comparing the responses between different age groups (Table 3), those between the ages of 55 and 64 years were least concerned about contracting the COVID-19 virus. Most age groups did not have negative experiences with previous vaccines, whereas those who were 65+ years old reported that they did. More 65+ year old respondents agreed that historical mistreatment of Black patients made them concerned about the vaccine compared to individuals between 55–64 years old, who were least concerned. Those 65 years and older were more concerned about the side effects of the COVID-19 vaccine than younger age groups. Individuals older than 65 years showed a trend towards agreeing more about the difficulty in traveling to vaccination sites compared to other individuals. Individuals aged between 35 and 54 years rtended to be most concerned about missing work. Those between 35 and 44 years showed a trend towards being most concerned about not finding childcare. Those 65 years and older tended to be more hesitant about receiving the vaccine due to religious beliefs.

When comparing the responses between race and ethnicity (Table 4), both racial categories disagreed that the vaccine was developed too quickly. Still, “All other races” agreed more than White respondents that vaccines were developed too quickly. Both racial categories disagreed that their social circle is not receiving the vaccine; yet White agreed more than all other races. Both racial categories were told by an institution to get the vaccine; “All other races” agreed more than White. All other races were trending towards being more concerned about long-term side effects of the vaccine than White respondents. Both racial categories agreed that it was better for the social circle and community for them to receive the vaccine, but Whites trended towards agreeing more than all other races.

When comparing the responses between provider types (Table 5), nurse practitioners and clinicians in training were more concerned about the long-term side effects of the vaccine than physicians. Physicians and nurse practitioners were more likely to believe that there was adequate testing of vaccines, but clinicians in training showed a trend towards being less confident about the testing.

When comparing the responses between family medicine specialties and non-family medicine specialties (Table 6), more family medicine clinicians agreed that the vaccine had adequate testing compared to non-family medicine clinicians. More non-family medicine clinicians had negative experiences with previous vaccines than family medicine clinicians. Trends showed that more non-family medicine clinicians were told by their social circle to receive the vaccine than family medicine clinicians. In addition, another trend showed that more family medicine clinicians agreed that their social circle was not receiving the vaccine than non-family medicine clinicians.

When comparing the responses between primary care specialties and non-primary care specialties (Table 7), more non-primary care HCWs had negative experiences with previous vaccines compared to primary care HCWs. A trend showed that more non-primary care HCWs were told by their social circle to get the vaccine than primary care HCWs.

### 3.3. Major Factors for Recommending a COVID-19 Vaccine to a Patient

Based on frequency, the two most important factors for recommending a COVID-19 vaccine to a patient were: efficacy of the vaccine, and current exposure to patients with active COVID-19 infection and risk of virus spread. These two factors accounted for 74% of all responses, as presented in Table 8.

### 3.4. Attitudes and Beliefs of HCWs Unlikely to Receive a COVID-19 Vaccine

Out of all of the participants, only five HCWs said they were unlikely to receive a COVID-19 vaccine (“If given the opportunity to take a COVID-19 vaccine, how likely is it that you would get the vaccine/shot?”: “Definitely will not”, N = 4; “Very unlikely”, N = 1). Those who selected “Definitely will not” selected that “I am NOT planning on getting vaccinated” on the survey. The HCW who selected “Very unlikely” selected “Yes, I have received the first dose of a two dose COVID-19 vaccine (i.e., Pfizer-BioNTech, Moderna)”. All five of these HCWs selected that (1) the COVID-19 vaccines did not have adequate testing and results, (2) the vaccines were developed and tested too quickly, and (3) they were concerned about the side effects and long-term effects of the COVID-19 vaccines. When asked “Given what you currently know, how likely would you recommend a COVID-19 vaccine when it becomes available to the patients you consult or treat?”, three HCWs selected “Somewhat likely”, one HCW selected “Not too likely”, and one HCW selected “Not at all likely”. The demographic characteristics of these five HCWs varied among gender, age, ethnicity, provider type, and specialty. The ranking of the most important factors for recommending a COVID-19 vaccine varied among these five HCWs.

## 4. Discussion

Previous studies have utilized certain HCWs’ views as predictors of their intent to get vaccinated and to recommend vaccination to high-risk patients [36]. In addition, past studies have demonstrated that the pivotal role of HCWs as sources of information has had a positive impact on vaccination attitudes and promoting successful herd immunity [37,38]. Collectively, this study characterizes HCWs’ attitudes toward the COVID-19 virus and vaccinations.

The impact of this study is strengthened by the 29-item opinions based on previous literature. This 29-item approach allowed our conclusions to support or contrast findings in other studies. It also allowed us to look at many different attitudes and concerns regarding both the COVID-19 virus and vaccines, which allowed us to find a diversity of associations. In addition, this study specifically focused on HCWs, a demographic that is less studied when it comes to vaccine hesitancy, and HCWs of diverse provider types [9,10,11]. Finally, the data was collected among clinics and healthcare institutions in four counties in Michigan, which increases the generalizability of our data. 

### 4.1. Gender Disparities across Vaccinations 

The study cohort presented a clear dichotomy between the male and female genders on their view of the COVID-19 vaccine: female respondents were significantly more concerned about being infected by the virus than male respondents (*p* = 0.011). Elevated concerns may be associated with increased vaccine uptake. A study on gender gaps in COVID-19 vaccine research by Vassallo et al. mentioned that the majority of the healthcare and hospital population are women, and when vaccines were rolled out on a risk-based prioritization, more women than men received the COVID-19 vaccine [39]. 

### 4.2. Variations in Vaccine Views and Hesitancy across Different Age Demographics 

Our study demonstrated statistically significant differences among those in the 55–64 years age range and the 65+ years age range with respect to vaccine views and hesitancy. The 55–64 years range was the least concerned about self-infection as compared to all other age ranges. Those in the 65+ age range reported they had negative experiences with previous vaccines, shared concerns over the historical mistreatment of those identifying as Black, and were more concerned about the side effects of the COVID-19 vaccines compared to other age groups. The results of this study differ from the results from Shih et al., which found more of a relationship between risk perception and vaccine acceptance for those aged 18–23 years compared to those older than 56 years [9]. Those in the younger age range were more likely to show vaccine hesitancy than those that were older (*p* = 0.0037). However, our study focuses on HCWs who are 25 years of age and older, and thus the data reveals the vaccine perception and acceptance in adult HCWs, which is the first report to our knowledge in this target population. 

### 4.3. Race, Ethnicity, and Disparities in Rates of Vaccination among HCWs 

The non-White races, which included Black, Asian, native Hawaiian, or other Pacific Islander, and other racial groups were more likely to agree that the vaccine was developed too quickly and that they were instructed to receive the vaccine. In contrast, people identifying as White were more likely to agree that their social circle was not yet receiving the vaccine.

Racial and ethnic disparities exist in all areas of medicine, and this is no different in vaccination acceptance and vaccine course completion. Previous research has shown that certain racial and ethnic groups are traditionally less likely to be vaccinated when compared to White populations within the United States. Li et al. showed lower H1N1 vaccination rates among non-Hispanic Black and American Indian/Alaskan native populations enrolled in Medicaid when compared to White populations enrolled in Medicaid across the United States [40]. However, the study also showed higher rates of H1N1 vaccination among Hispanic and Asian/Pacific Islander populations enrolled in Medicaid when compared to White populations enrolled in Medicaid. Other studies conducted regarding vaccination disparities regarding HPV and influenza suggest that a lack of vaccination of certain racial and ethnic populations, when compared to White populations, are due a lack of education and awareness of the vaccinations and could be mediated through proper education [41,42]. 

The similarities seen in the data collected between the two racial categories within this study could be due to similarities in the location of the surveyed individuals and the high baseline of agreement among physicians for vaccine acceptance. As will be discussed at length in the following section, the surveyed physicians had high levels of acceptance towards the COVID-19 vaccine and were willing to receive the vaccine upon availability [34]. Considering that 68.9% of respondents in this study were physicians when compared to the other provider types, the traditionally high rates of vaccine acceptance amongst physicians could have outweighed any vaccination beliefs that resulted from traditional racial disparities. Therefore, the traditional disparities and differences amongst races and ethnicities when discussing vaccine acceptance may be outweighed by the effect of medical education, suggesting medical education promotes higher rates of trust in medical research.

### 4.4. HCW Provider Type and Varying Levels of Trust in Vaccine Research 

When analyzing the cohort by HCW provider type, nurse practitioners and other clinicians-in-training were more concerned about long-term side effects when compared to their physician counterparts. These trends indicate a lower level of trust in the vaccine research amongst those in the nursing profession when compared to the other analyzed provider types. 

Previous studies have suggested that those within the nursing profession tend to show suboptimal uptake of vaccines [30,43,44]. Additional studies, when focusing specifically on the creation and distribution of the COVID-19 vaccines, further confirm this finding. Kwok et al. showed, through a survey of 1205 nurses, that less than two-thirds of surveyed nurses would be willing and ready to receive the COVID-19 vaccine when available [30]. Despite the urgency of the active pandemic, willingness to get vaccinated still fell below herd immunity levels. Wang et al. also showed that surveyed nurses in Hong Kong displayed lower than optimal acceptance of both influenza and COVID-19 vaccines. In addition, a study by Lee et al. found that HCWs with the most patient contact, such as nurses and aides, often showcased lower rates of vaccination compared to physicians [45]. The study emphasized the necessity of effective campaign strategies in order to increase vaccine uptake among nursing staff. The listed studies emphasized the connection between higher education levels regarding the vaccines and higher levels of peer acceptance of the vaccines with vaccine acceptance. 

The significant trend displayed in the HCWs analyzed within this study, with physicians tending to have lower levels of vaccine hesitancy than nurse practitioners and other clinicians-in-training, also matches research with vaccine acceptance amongst physicians. Day et al. surveyed physicians and showed that most of the physicians were willing to receive the COVID-19 vaccine immediately upon availability and felt comfortable easing patients’ concerns regarding the vaccine [46]. Additionally, a cross-sectional study of HCW attitudes towards vaccine acceptance showed that physicians had the highest level of vaccine acceptance rates when compared to nurses, allied healthcare professionals, and Master’s level clinicians [47]. 

The trends of vaccine acceptance versus hesitancy amongst the provider types could be directly tied to varying levels of research education and the direct impact of research amongst the varying healthcare professions. Therefore, as the length of time in educational years of training increases, it could be assumed that the healthcare provider would have higher levels of acceptance towards medical research, research scientists, and, therefore, vaccinations. Additionally, it could be assumed that not only the length of educational years, but the length of educational years that focus on the natural sciences rather than solely on clinical skills for patient care could lead to higher levels of trust in research and vaccine acceptance. 

### 4.5. Views on Vaccine Safety and Efficacy Based on Medical Specialty

Even among clinicians with similar levels of baseline training in the sciences, there can be different viewpoints among various subspecialties. A significant number of family medicine physicians reported that they believed the COVID-19 vaccine had adequate testing compared to non-family medicine physicians (*p* = 0.028). A study by Ofei-Dodoo et al. found that a great proportion of family physicians intended to be vaccinated with an approved COVID-19 vaccine. The study surveyed 307 practicing family physicians and unveiled a significantly greater rate of family medicine physicians reporting their intention to be COVID-19 vaccinated than the rate of family medicine physicians who were hesitant to receive the COVID-19 vaccines [48]. It could be surmised that family medicine specialists are more attuned to vaccinations due to a greater focus on preventative measures than other specialties.

When stratifying the data based on primary care specialties, our own study revealed a statistically significant increase in non-primary care physicians who have a prior negative vaccination experience compared to primary care clinicians (*p* < 0.008). A national survey conducted from February to March 2021 surveying pediatric primary care professionals (N = 1047) found that most primary care physicians (83%) supported COVID-19 vaccination mandates for HCWs, especially among individuals who perceived HCWs to be at a greater risk of contracting the virus [49].

### 4.6. Top Factors for Recommending the COVID-19 Vaccine 

As showcased in Table 8, the four most important factors for an HCW to recommend a COVID-19 vaccine to their patients include, in descending order of importance: (1) efficacy of vaccine, (2) current exposure to patients with active COVID-19 infection and concern of risk of virus spread, (3) safety of vaccine and long-term follow-up, and (4) recommendation of vaccine by healthcare authorities. These four factors accounted for 97% of all responses. Coming in a close fourth place is “recommendation of vaccine by healthcare authorities”. Less important factors for consideration include the following: duration of protection by vaccine, incidence of major and minor adverse effects, and recommendation of vaccine by political officials. An investigation conducted by researchers at the University of Nevada Reno School of Medicine, in collaboration with Immunize Nevada and the Nevada Department of Health and Human Services, discussed the increased rates of likelihood to receive the COVID-19 vaccine among those who previously and currently receive influenza vaccines. Additionally, the Nevada study discovered that the most important factors people consider when receiving the vaccine include efficacy, safety, and adverse effects of the COVID-19 vaccine. Overall, 77% of HCWs stated they would likely receive the vaccine, and 83% declared they were likely to recommend the vaccine. The majority of HCWs recommending vaccination for their patients and themselves were more likely attending physicians, and less likely interns and residents/fellows [50].

An additional study, completed by Ofei-Dodoo et al., supported our own findings, as physicians also reported the most important reasons to become vaccinated as being the following: preventing COVID-19 infection, protecting themselves and their families, friends, and communities, and inspiring confidence in others to become vaccinated [48]. Finally, Di Giuseppe et al. and Verger et al. also identified vaccine safety as a critical factor for vaccine acceptance among HCWs in their respective studies [51,52]. These studies showcased the importance of HCWs’ influence on the general population regarding vaccination, and the importance of presenting the safety and efficacy of elective vaccines in garnering higher vaccination rates.

### 4.7. Limitations 

Our data revealed that, despite hesitancy and negative beliefs regarding the COVID-19 vaccines among HCWs, all except two of the HCWs stated that they follow the CDC guidelines and recommend the vaccines to their patients. Interestingly, although five HCWs selected that they would not get the vaccine, only two of those HCWs noted that they will not recommend the COVID-19 vaccine to their patient. This may be due in part to social desirability bias. Alternatively, HCWs against vaccination may have been the non-responders of this study, since only 30.3% of the targeted population accepted and responded to this survey. We acknowledge that this study does not detail information regarding non-responders. We were not able to quantify the reasons for this study’s non-responders due to the parameters set by the IRB guidelines. Additionally, our data showed that physicians, who are more likely to support vaccination, were overrepresented compared to other healthcare provider types [53]. Finally, there may have been an association with acceptance of vaccines and the years of experience working in healthcare which was not studied in this survey.

The data collection timeline was spread out, from July 2021 until the end of November 2021. Many pandemic-related factors, such as the various new variants, CDC recommendation changes, and COVID-19 infection surges, may have impacted HCW opinions depending on when the participant took the survey [54]. We did not further report changes in attitudes at different time points due to the limited sample size, that was not sufficient to reach statistically significant conclusions. 

The generalizability of this study’s findings is limited since it took place in one state in the United States. Therefore, other states in the United States or countries with different demographics, such as ethnicity, may yield alternative findings. Our study sampled specific healthcare institutions because the selected institutions served the majority of each given county population. We do acknowledge that we did not survey all institutions in the area, which may affect the margin of error of this study. In addition, this limited sample size gave rise to many meaningful trends. Further studies using a larger sample size could cross-validate the relationships and associations found in our study.

## 5. Conclusions

HCWs play an important role in an individual’s vaccination decision. Therefore, it is crucial that HCWs view vaccination positively during public health crises where mass vaccination plays a pivotal role in the eradication of a disease, such as during the COVID-19 pandemic. This study demonstrated how demographic characteristics, such as gender, age, race, provider type, and medical specialty among HCWs in Michigan show statistically significant differences regarding the COVID-19 virus and vaccines acceptance. 

In addition, this study illustrated many meaningful trends for future investigations. For instance, when it came to gender, female respondents in our study were more likely to agree that the CDC recommendations were effective against the virus and were more concerned about the long-term effects of the vaccines as compared to the male respondents. A meaningful trend in provider type showed that nurse practitioners and physicians were more likely to believe that adequate time was spent researching the vaccine when compared to clinicians-in-training. Additional trends showed that more non-family medicine physicians were told by their social circle to receive the vaccine and generally more family medicine physicians confessed that their social circle is not receiving the vaccine. Exploring and understanding these trends can help to refine public health measures to increase vaccination among HCWs and their community. Ultimately, while most HCWs received and recommended the COVID-19 virus, this study demonstrated the importance that factors such as the efficacy and safety of the vaccine and the risk of virus spread play when it comes to recommending a COVID-19 vaccine to a patient. 

The data in this study demonstrated the importance of developing and implementing educational interventions for HCWs who are more likely to have negative attitudes, with the goal of decreasing HCWs’ hesitancy and increasing vaccination rates in their communities. An effective educational intervention based on the data reported in this study is a crucial future step, as the acceptance of a COVID-19 vaccine by HCWs is of the utmost importance for (1) protecting patient safety, (2) improving the occupational health and well-being of the HCWs themselves, (3) role modeling and public trust, as HCWs’ attitudes and behaviors towards vaccination can significantly impact public perceptions and acceptance of a COVID-19 vaccine, and (4) improving the outbreak response and preparedness of these frontline workers. Understanding the attitudes and beliefs of HCWs towards vaccinations and applying these results to build interventions to address vaccination hesitancy and other public health concerns is a critical tool in mitigating the spread of such diseases within the community and public.

## Figures and Tables

**Figure 1 vaccines-11-01105-f001:**
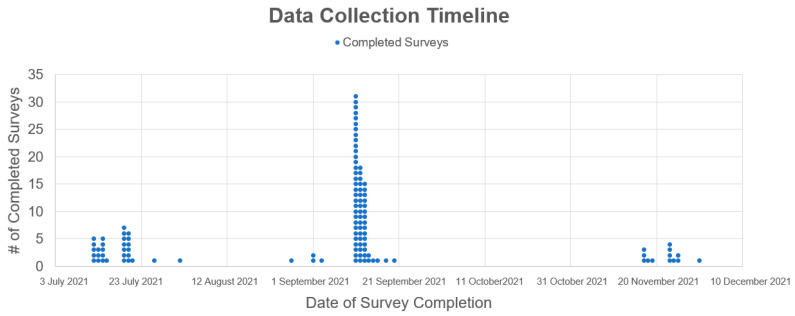
Data collection timeline. Dates and quantity of completed survey during data collection timeline.

**Figure 2 vaccines-11-01105-f002:**
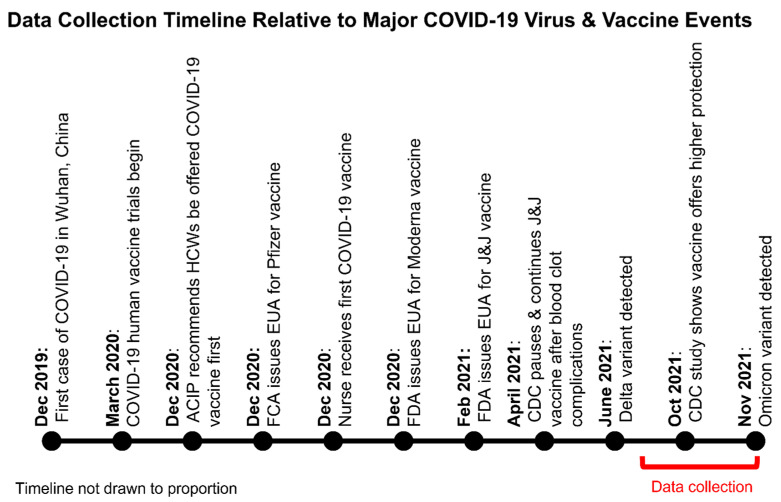
Major COVID-19 events during data collection. Data collection period relative to major COVID-19 pandemic events reported by Centers for Disease Control and Prevention [13].

**Table 1 vaccines-11-01105-t001:** Participant demographics (N = 120).

	N%
Gender	
Female	64 (53.8%)
Age	
25–34 years	23 (19.2%)
35–44 years	34 (28.3%)
45–54 years	22 (18.3%)
55–64 years	31 (25.8%)
65+ years	10 (8.3%)
Race/Ethnicity	
White	98 (82.4%)
All other races	21 (17.6%)
Provider types	
Physicians	82 (68.9%)
Nurse practitioners	20 (16.8%)
Clinicians in training	17 (14.3%)
Family medicine	
Family medicine	45 (37.8%)
Other specialties	74 (62.2%)
Primary care	
Primary care	74 (62.2%)
Other specialties	45 (37.8%)
Flu shot last year	
Yes	115 (96.6%)
Flu shot this year	
Yes	112 (96.6%)
Tested for COVID-19	
Yes	75 (61.0%)
Getting a COVID-19 vaccine	
Yes (Somewhat likely, very likely, definitely will, already received vaccine)	117 (95.9%)
Recommended the COVID-19 vaccine	
Yes (Extremely likely, very likely, somewhat likely)	114 (98.3%)

**Table 2 vaccines-11-01105-t002:** Mean item scores by respondent gender (N = 119).

Abbreviated Item Content	Female N = 64	Male N = 55	*t*-Test	*p*-Value
Concern about contracting COVID-19 virus	1.52	1.09	2.59	**0.011**
Belief in CDC recommendations	1.88	1.65	1.90	0.061
Concern about long term effects of vaccine	0.67	0.42	1.71	0.09
Hard to know whom to trust for info	0.33	0.6	−1.78	0.077

Bold values denote statistical significance at the *p* < 0.05 level.

**Table 3 vaccines-11-01105-t003:** Mean item scores by respondent age (N = 120).

Abbreviated Item Content	25–34 N = 23	35–44 N = 34	45–54 N = 22	55–64 N = 31	65 + N = 10	F	*p*-Value
Concern about contracting COVID-19 virus	1.43	1.53	1.41	0.87	1.4	2.661	**0.036**
Concern about side effects of vaccine	0.3	0.62	0.64	0.19	1	2.982	**0.022**
Negative experience with previous vaccine	0	0.12	0.18	0.06	0.8	5.112	**<0.001**
Historical mistreatment with Black	0.26	0.15	0.32	0.06	0.7	2.824	**0.028**
Difficult to travel	0	0	0.05	0	0.2	2.338	0.059
Worried about missing work	0.43	0.59	0.55	0.06	0.2	2.32	0.061
Find childcare	1.04	1.65	1.27	1.1	1	2.403	0.054
Religion	0	0.06	0.18	0.03	0.4	2.28	0.065

Bold values denote statistical significance at the *p* < 0.05 level.

**Table 4 vaccines-11-01105-t004:** Mean item scores by race/ethnicity (N = 119).

Abbreviated Item Content	White N = 98	All Other N = 21	*t*-Test	*p*-Value
Vaccine developed too quickly	0.25	0.71	−2.41	**0.024**
Concern about long term effects of vaccine	0.5	0.9	−2.03	0.051
Social circle is not receiving vaccine	0.6	0.24	2.46	**0.017**
Institution told me to get vaccine	1.6	1.9	−2.43	**0.019**
Better for social circle and community to get vaccine	1.9	1.52	1.92	0.068

Bold values denote statistical significance at the *p* < 0.05 level.

**Table 5 vaccines-11-01105-t005:** Mean item scores by provider type (N = 119).

Abbreviated Item Content	Physicians N = 82	Nurse Practitioners N = 20	Clinicians in Training N = 17	F	*p*-Value
Adequate testing of vaccine	1.76	1.85	1.41	2.858	0.061
Concern about long term effects of vaccine	0.41	0.8	0.94	4.192	**0.017**

Bold values denote statistical significance at the *p* < 0.05 level.

**Table 6 vaccines-11-01105-t006:** Mean item scores by specialty (family medicine) (N = 119).

Abbreviated Item Content	FM N = 45	NFM N = 74	*t*-Test	*p*-Value
Adequate testing of vaccine	1.87	1.64	2.23	**0.028**
Negative experience with previous vaccine	0.04	0.22	−2.02	**0.046**
Social circle told me to get vaccine	1.29	1.61	−1.87	0.066
Social circle is not receiving vaccine	0.71	0.43	1.70	0.094

Bold values denote statistical significance at the *p* < 0.05 level. FM, family medicine; NFM, non-family medicine.

**Table 7 vaccines-11-01105-t007:** Mean item scores by specialty (primary care) (N = 119).

Abbreviated Item Content	PC N = 74	NPC N = 45	*t*-Test	*p*-Value
Negative experience with previous vaccine	0.03	0.36	−2.78	**0.008**
Social circle told me to get vaccine	1.38	1.67	−1.87	0.065

Bold values denote statistical significance at the *p* < 0.05 level. PC, primary care; NPC, non-primary care.

**Table 8 vaccines-11-01105-t008:** Most important factors for recommending a COVID-19 vaccine to a patient (N = 111).

Factor for COVID-19 Vaccine Recommendation	Frequency (N)
Efficacy of vaccine	47
Safety of vaccine and long-term follow-up	14
Duration of protection by vaccine	0
Incidence of major and minor adverse effects	2
Recommendation of vaccine by political officials	1
Recommendation of vaccine by healthcare authorities	12
Current exposure to patients with active COVID-19 infection and risk of virus spread	35
Total	111

## Data Availability

The datasets generated and/or analyzed during the current study are not publicly available due to the nature of the Institutional Review Board protocol but are available from the corresponding author on reasonable request.

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
