# Peer review of "COVID-19 Virus and Vaccination Attitudes among Healthcare Workers in Michigan: A Cross-Sectional Study"

_vaccines, 2023, doi:10.3390/vaccines11061105_

Round 1
Reviewer 1 Report
This is a descriptive study of questionnaire responses regarding COVID19 vaccination from 120 healthcare workers in primary care from Michigan taken from July to Nov 2021. They found that over 95% of respondents had received the vaccination and that there were some differences in attitudes between different genders, ethnicities, age groups, and specialties of the respondents. The study is limited by the relative small sample size and time the survey was given.
Specific comments:
1. In the abstract please clarify what “adequate testing” refers to. Also in the abstract please indicate the results of the question most directly related to vaccine hesitancy, for example the result for the question on whether the respondent would recommend a COVID vaccine.
For all the tables of data please indicate the range of possible scores (i.e. 0-3) and whether a higher score indicated agreement or disagreement.
3. For Table 1 – please indicate the percentage of yes answers to these questions within each category of age group, gender, professional role, and specialty.
4. Results and discussion – please indicate the attitudes of the 5 respondents that said they would not get the vaccine – was there a predominant reason or set of reasons for this?
Author Response
Thank you for your feedback. We have addressed your comments below:
- In the abstract please clarify what “adequate testing” refers to. Also in the abstract please indicate the results of the question most directly related to vaccine hesitancy, for example the result for the question on whether the respondent would recommend a COVID vaccine.
Response: We have changed “adequate testing” to “adequate clinical trial testing” in line 25 to clarify this point. We have added more results to the abstract in line 18-21 regarding the number of HCWs who received and recommended a COVID-19 vaccine and the top three factors for recommending a COVID-19 vaccine, “Overall, most HCWs received (N=117, 95.9%) and recommended (N=114, 98.3%) a COVID-19 vaccine. The top three factors that HCWs cited for recommending a COVID-19 vaccine are: 1) Efficacy of the vaccine (N=47, 42.3%), 2) current exposure to patients with active COVID-19 infection and risk of virus spread (N=35, 31.5%), and 3) safety of vaccine and long-term follow-up (N=14, 12.6%).”
- For all the tables of data please indicate the range of possible scores (i.e., 0-3) and whether a higher score indicated agreement or disagreement.
Response: The range of possible scores is 0-2 (0=Disagree, 2=Agree). We have added the following under lines 154-158), “Tables 2-7 represents mean item scores of HCWs regarding concerns and beliefs of the COVID-19 virus and vaccines. A value closer to ‘0’ indicates that the demographic group disagreed more and a value closer to ‘2’ indicates that the group agreed more with the statements abbreviated in each column.”
- For Table 1 – please indicate the percentage of yes answers to these questions within each category of age group, gender, professional role, and specialty.
Response: Thank you for your feedback. We are concerned that Table 1 already contains a lot of information. Adding the percentage of “yes” answers to the questions within each category of age group, gender, professional role, and specialty will expand the table size at least three times. We would like the reviewer to please reconsider how valuable to the manuscript they believe adding the additional information will be and to allow us to use Table 1 as it is.
- Results and discussion – please indicate the attitudes of the 5 respondents that said they would not get the vaccine – was there a predominant reason or set of reasons for this?
Response: We have added more details on the attitudes of the 5 respondents that said they did not get the vaccines under the “Results” section, lines 219-235 “Out of all participants, only five HCWs said they were unlikely to receive a COVID-19 vaccine (“If given the opportunity to take a COVID-19 vaccine, how likely is it that you would get the vaccine/shot?”; “Definitely will not,” N=4; “Very unlikely,” N=1). Those who selected “Definitely will not” selected that “I am NOT planning on getting vaccinated” on the survey. The HCW who selected “Very unlikely” selected “Yes, I have received the first dose of a two dose COVID-19 vaccine (i.e. Pfizer-BioNTech, Moderna).” All five of these HCWs selected that (1) the COVID-19 vaccines did not have adequate testing and results, (2) the vaccines were developed and tested too quickly, and (3) they were concerned about the side effects and long-term effects of the COVID-19 vaccines. When asked “Given what you currently know, how likely would you recommend a COVID-19 vaccine when it becomes available to the patients you consult or treat?,” three HCWs selected “Somewhat likely,” one HCW selected “Not too likely,” and one HCW selected “Not at all likely.” The demographical characteristics of these 5 HCWs varied among gender, age, ethnicity, provider type, and specialty. The ranking of most important factor for recommending a COVID-19 vaccine varied among these five HCWs.”
Reviewer 2 Report
Nice study and a survey that looks to have covered many aspects associated with the research. My questions are related to:
- Did you look at years of experience working in healthcare? This may be associated with acceptance of vaccines and the role of educating patients on vaccination.
- You touched upon this on the Discussion, but how many patients treated on average for each of the types of HCWs? How many were COVID-19 infected? These could have been placed into the survey.
- It could be surmised that family medicine is far more attuned to vaccinations due to a greater focus on preventive measures than other types of healthcare.
- What was the willingness of HCWs to educate their patients on the importance of vaccination and if not, why not?
- If those HCWs had bad experiences with vaccinations, why?
I believe that these need to be mentioned in some capacity in the paper.
Author Response
Thank you for your feedback. We have addressed your comments below:
- Did you look at years of experience working in healthcare? This may be associated with acceptance of vaccines and the role of educating patients on vaccination.
Response: We unfortunately did not assess this in our original survey. We have added this as a limitation to the study under the Discussion section, lines 401-403, “Finally, there may have been an association with acceptance of vaccines and the years of experience working in healthcare which was not studied in this survey.”.
- You touched upon this on the Discussion, but how many patients treated on average for each of the types of HCWs? How many were COVID-19 infected? These could have been placed into the survey.
Response: Thank you for your feedback. In our survey, we did not ask HCWs about how many patients they treated or how many of their patients were infected by COVID-19 so we unfortunately cannot elaborate on these points in our Discussion.
- It could be surmised that family medicine is far more attuned to vaccinations due to a greater focus on preventive measures than other types of healthcare.
Response: Yes, we agree with this statement and have included it under our Discussion in lines 351-353 and added the appropriate reference.
- What was the willingness of HCWs to educate their patients on the importance of vaccination and if not, why not?
Response: Although we analyzed the likelihood of recommending a COVID-19 vaccine to patients and major factors for recommending a COVID-19 vaccine to a patient, we did not assess the willingness to educate their patients on the importance of vaccination due to the limited length of the survey.
- If those HCWs had bad experiences with vaccinations, why?
Response: We asked participants if they ‘AGREE,’ ‘DISAGREE,’ or ‘UNSURE’ for the following statements, “I have previously had a negative experience with other vaccines.” However, we did not have a free response section on the survey for participants to elaborate on their selection/ We did not learn about the bad experiences until after analyzing the survey data. This would be an interesting topic to explore in future studies.
Reviewer 3 Report
The authors proposed to determine healthcare workers’ (HCW) attitudes regarding the COVID-19 vaccination and reasons for vaccine hesitancy. The study assessed, in a cross-sectional study in Michigan USA, HCWs’ attitude, especially demographics and personal profiles, using tipping scale questions (with a validated 29-item questionnaire). If we already got knowledge on COVID-19 vaccination and attitudes of HCWs towards it, complementary data are still of interest to help focusing educational efforts on hesitant HCWs regarding actual factors to address.
Hence, this study is complementary to previous ones, cited in introduction and discussion, giving details about HCWs’ attitudes toward COVID vaccination. Previous literature presented the findings of predictors of the intent of vaccinating self and of recommending vaccination to high-risk patients (ref 27) or demonstrating the pivotal role of HCWs on vaccination attitudes and promoting successful herd immunity (references 28, 29). Here details about the profile of the HCWs are specifically studies, and of interest for the readers.
This work is worthy due to its potential help to decision making and public health policies toward HCWs education and efforts to promote them to be the first ambassadors of vaccination, and though requiring first to convince them. The major strength is the specific robust questionnaire validated in the literature that could help addressing the diversity of associations in COVID attitudes in HCWs, and to study the impact of demographic in vaccine hesitancy.
The major limit, in my opinion, is the profile of the HCWs that replied to the study; 96% were vaccinated, and 98% recommended the vaccination. Though the bias of response seems in 2021 to be very important, and the assessment of vaccine hesitancy is hard to study due to the probably overestimation of the pro vaccine in the population of respondents. The main point is that the HCW against vaccination must have not answered (only ¼ of the target population did participate to the study), that we could underline also with the overrepresentation of physicians, known as more pro vaccination than paramedics.
The second limit is the generalizability of the findings regarding the study population coming from one USA State, and the study of some demographics such as ethnicity that other countries can’t study or use as an intervention tool (like in several European countries). This must be discussed more in the discussion.
Minor comments:
The results could be summarized and the tables could be gathered and some could be removed, in my opinion.
Idem for the discussion, should be synthesized.
very good
must be summarized
Author Response
Thank you for your feedback. We have addressed your comments below:
- The major limit, in my opinion, is the profile of the HCWs that replied to the study; 96% were vaccinated, and 98% recommended the vaccination. Though the bias of response seems in 2021 to be very important, and the assessment of vaccine hesitancy is hard to study due to the probable overestimation of the pro vaccine in the population of respondents. The main point is that the HCW against vaccination must have not answered (only ¼ of the target population did participate to the study), that we could underline also with the overrepresentation of physicians, known as more pro vaccination, than paramedics.
Response: We agree with the limits. We have elaborated in these points under the “Limitations” section under lines 395-401, “Alternatively, HCWs against vaccination may have been the non-responders of this study since only 30.3% of the targeted population accepted and responded to this survey. We also acknowledge that this study does not detail information regarding non-responders. We were not able to quantify the reasons for this study’s non-responders due to the parameters set by the IRB guidelines. Additionally, our data showed that physicians, who are more likely to support vaccination, were overrepresented compared to other healthcare provider types (44).”
2. The second limit is the generalizability of the findings regarding the study population coming from one USA State, and the study of some demographics such as ethnicity that other countries can’t study or use as an intervention tool (like in several European countries). This must be discussed more in the discussion.
Response: We agree with the limits, and this has been added to the discussion section. We have elaborated these points in lines 410-413, “The generalizability of this study’s findings is limited since it took place in one state in the United States. Therefore, other states in the United States or countries with different demographics, such as ethnicity, may yield alternative findings and, consequently, the results of this study cannot be used as a tool to guide HCW-targeted interventions.”
3. The results could be summarized, and the tables could be gathered and some could be removed, in my opinion. Idem for the discussion, should be synthesized.
Response: Thank you for your suggestions. We have done our best to synthesize the results and discussion section. If the respective reviewer can suggest directed changes (i.e. which tables, discussion topics) we will consider revision. Thank you for your comments. Your feedback is greatly appreciated.